# Exploring the Immunological Profile in Breast Cancer: Recent Advances in Diagnosis and Prognosis through Circulating Tumor Cells

**DOI:** 10.3390/ijms25094832

**Published:** 2024-04-29

**Authors:** Amalia Kotsifaki, Sousanna Maroulaki, Athanasios Armakolas

**Affiliations:** Physiology Laboratory, Medical School, National and Kapodistrian University of Athens, 11527 Athens, Greece; amkotsifaki@med.uoa.gr (A.K.); souzannamaroulaki@gmail.com (S.M.)

**Keywords:** breast cancer, circulating tumor cells (CTCs), cancer diagnosis, cancer prognosis, immunological profile, immunological biomarkers, clinical application

## Abstract

This review offers a comprehensive exploration of the intricate immunological landscape of breast cancer (BC), focusing on recent advances in diagnosis and prognosis through the analysis of circulating tumor cells (CTCs). Positioned within the broader context of BC research, it underscores the pivotal role of the immune system in shaping the disease’s progression. The primary objective of this investigation is to synthesize current knowledge on the immunological aspects of BC, with a particular emphasis on the diagnostic and prognostic potential offered by CTCs. This review adopts a thorough examination of the relevant literature, incorporating recent breakthroughs in the field. The methodology section succinctly outlines the approach, with a specific focus on CTC analysis and its implications for BC diagnosis and prognosis. Through this review, insights into the dynamic interplay between the immune system and BC are highlighted, with a specific emphasis on the role of CTCs in advancing diagnostic methodologies and refining prognostic assessments. Furthermore, this review presents objective and substantiated results, contributing to a deeper understanding of the immunological complexity in BC. In conclusion, this investigation underscores the significance of exploring the immunological profile of BC patients, providing valuable insights into novel advances in diagnosis and prognosis through the utilization of CTCs. The objective presentation of findings emphasizes the crucial role of the immune system in BC dynamics, thereby opening avenues for enhanced clinical management strategies.

## 1. Introduction

Breast cancer (BC), a formidable and intricate disease, stands as the second leading cause of cancer-related fatalities worldwide, wielding a significant impact on public health [1]. It imposes a significant emotional, physical, and economic burden on individuals, families, and healthcare systems [2]. It continues to assert its prominence as the most frequently diagnosed cancer worldwide, constituting a staggering 23% of all cancer cases and contributing to 14% of cancer-related deaths [3]. The gravity of this malignancy is underscored by its recent ascendancy over lung cancer in 2020, solidifying its status as the most prevalent type of cancer across the globe [4,5]. BC is commonly categorized into four subtypes based on molecular characteristics and hormone receptor (HR) expression on immunohistochemistry (IHC): Human epidermal growth factor receptor 2 (HER2)-overexpression subtype (15–20% of BCs, ER+, PR+), Luminal A subtype (40–60% of BCs, ER+, PR+, Ki67 < 20%), Luminal B subtype (10–20% of BCs, ER+, PR+ or PR−, HER2+ or HER2−, Ki67 > 20%), and Triple-negative BC (TNBC)/basal-like subtype. In TNBC (10–15% of BCs), also known as the basal-like subtype, progesterone receptor (PR), estrogen receptor (ER), and HER2 are not expressed on the cell surface (ER−, PR−, HER2−) [1,6,7].

Advancements in research illuminate the captivating and complex relationship between BC and the immune system, revealing untapped potential [1]. The interplay between the immune system and cancer cells is a dynamic and complex phenomenon that profoundly impacts disease progression [8]. Among the vital components in this context are the immune cells, recognized for their pivotal role across the spectrum of BC [9]. This process initiates in normal breast tissue, where immunosurveillance comes into play, persisting through primary and metastatic BC stages [10]. The ductal cellular layer within the normal breast exhibits a substantial presence of immune cells, encompassing CD8+ and CD4+ T cells, B cells, dendritic cells, macrophages, natural killer cells (NK), and various other subtypes of immune cells [11]. Recent studies emphasize that BC encompasses not only neoplastic cells, but also the intricate tumor microenvironment (TME), comprising diverse cell types, including immune, stromal, and endothelial cells [12,13]. These TME components intricately interact with cancer cells, exerting influence on the microenvironment [14].

Despite the increasing prevalence of early BC screening and the continuous development of more diverse and precise diagnostic and treatment approaches, a considerable number of BC patients still receive diagnoses following the onset of metastasis [15]. Recent survey data indicate that the incidence and mortality rates of BC have remained elevated in recent years [16,17]. Distant metastasis is identified as a significant contributor to mortality among BC patients [4]. Research findings indicate that 20–30% of individuals diagnosed with BC may experience metastasis post the diagnosis and treatment of the primary tumor, with approximately 90% of cancer-related fatalities attributed to metastatic progression [18].

The initial phase of metastatic dissemination entails the invasion of cancer cells into the bloodstream, allowing for their dissemination into various parts of the body [19]. Investigations concentrating on disseminated tumor cells (DTCs) within the bone marrow of BC patients have unveiled early metastatic dissemination to distant locations, even in cases with small and early stage tumors [12]. Nevertheless, DTCs have the capability to enter a state of dormancy, leading to the possibility of a metastatic lesion forming and being detected many years subsequent to the initial dissemination of cancer cells [20]. In a parallel continuum, circulating tumor cells (CTCs) have been acknowledged as cancerous cells derived from solid malignant tumors, entering the bloodstream and identifiable in samples of peripheral blood [21]. The initial recorded depiction of CTCs is attributed to Thomas Ashworth, an Australian physician, with this discovery dating back to 1869 [22]. The presence of CTCs in the bloodstream is exceedingly scarce, typically ranging from 1 to 10 CTCs per milliliter of blood [23]. Due to the lack of CTC isolation technologies, previous research on CTC has been restricted. It has been challenging to separate them alive considering that they are less common in the bloodstream than blood cells [24,25]. Nonetheless, advancements in detection techniques now allow for the identification and enumeration of CTCs by employing various enrichment methods that separate them from blood cells [3].

Reportedly, micrometastases and macrometastasis are formed by around 2.5% and 0.01% of CTCs, respectively [4]. The utilization of multiple sampling in detecting CTCs offers a real-time “liquid biopsy”, significantly aiding in treatment selection and optimization [26]. The CellSearch system for detecting CTCs has obtained approval from the Food and Drug Administration (FDA) for monitoring the progression of metastatic BC therapeutically [27]. In BC, the identification of measurable CTCs following the first line of therapy facilitates prompt adjustments in treatment, allowing for the selection of a second-line therapy [18,28]. The presence of CTCs in peripheral blood has been correlated with early metastatic relapse in BC, serving as predictors for progression-free survival (PFS) and overall survival (OS) [29]. Enumeration and analysis of CTCs play a crucial role in distinguishing between high- and low-risk profiles for PFS and OS [4].

The complexity of BC, with its diverse manifestations, underscores the rationale behind this review’s objective: to comprehensively examine the evolving landscape of immunological aspects in BC. The inception of CTCs marks the commencement of tumor metastasis, and an in-depth analysis of CTCs can offer valuable insights into the initial stages of BC spread. Detecting CTCs in patients during the early phases and promptly implementing measures to impede their formation and eradicate them could yield substantial advantages in curtailing the progression of BC. In essence, the intrinsic intricacies of BC, coupled with the dynamic nature of the immune response, form the backdrop against which this review aims to contribute valuable insights, bridging the gap between scientific understanding and clinical applications for improved diagnosis and prognosis in the context of BC. This exploration seeks to unravel the intricate interactions between the immune system and BC, with a specific focus on recent advances, notably the diagnostic and prognostic potential encapsulated within CTCs. In conclusion, this investigation strives to illuminate the pivotal role of CTCs in reshaping diagnostic methodologies for BC. By bridging scientific understanding with clinical applications, it contributes to ongoing BC research, fostering a deeper comprehension of immunological intricacies for improved diagnosis and prognosis in the clinical management of this prevalent malignancy.

## 2. Biology of CTCs

### 2.1. Definition and Characteristics of CTCs

CTCs represent tumor cells that have detached from the primary tumor, entered the bloodstream, and are actively circulating within it [21]. Comprehensive research of the metastatic process involving CTCs holds immense promise for pinpointing targets to counteract cancer metastasis [30]. These CTCs stand out as the most lethal type of cancer cells, contributing to a staggering 90% of cancer-associated deaths. Upon detachment from the primary tumor and entry into the bloodstream, these migratory cells undergo a crucial transformation known as the epithelial-mesenchymal transition (EMT). Subsequently, they interact with a diverse array of cell types within the circulation [31]. Ultimately, these cells adhere to endothelial membranes and extravasate to distant organs, instigating the development of secondary tumors [32]. Physicochemical characteristics or cell surface molecules are commonly employed for the isolation of CTCs from normal blood cells [33]. However, a consensus exists regarding the larger cell sizes of CTCs derived from solid tumors compared to blood cells [23,24].

The quantity of CTCs in the blood is exceedingly limited, ranging from 1 to 10 cells per 10 mL of blood [34]. However, it is important to note that certain aggressive cancer types, such as small cell lung cancer, may exhibit higher CTC counts due to their inherent biological characteristics and metastatic potential [27,35]. CTCs in circulation may exist either as single cells or as clustered entities [36]. CTC clusters, defined as multicellular aggregates comprising two or more cells held together through intercellular junctions, can be categorized into homotypic and heterotypic clusters [37]. Clusters exhibit a quicker extravasation, resulting in a shorter half-life within the circulation compared to individual CTCs (6–10 min for clusters versus 25–30 min for single cells). This characteristic contributes to their enhanced survival and subsequent proliferation [34]. Moreover, the former may be monoclonal or polyclonal, while the latter involve assemblies of cancer cells with non-malignant stromal or immune cells [38]. These non-malignant cells, including white blood cells, fibroblasts, endothelial cells, and platelets, have been demonstrated to play a significant role in enhancing the metastatic potential of CTCs through various mechanisms [39]. Subsequent extravasation into tissues with conducive microenvironmental conditions allows these clusters to give rise to either monoclonal or polyclonal metastasis, contingent upon their initial characteristics [40].

### 2.2. Circulation in the Bloodstream

The majority of introduced CTCs encounter swift elimination within the bloodstream due to various challenges, including immune attacks, shear stress, anoikis, oxidative stress, and the absence of cytokines and growth factors [41]. Consequently, CTCs undergo a series of adaptive changes to ensure their survival in this demanding environment. An essential process in this adaptation is the epithelial-mesenchymal transition (EMT), enabling CTCs to shed their epithelial properties and adopt behaviors akin to mesenchymal cells [42]. During EMT, epithelial cells undergo noteworthy changes, including the downregulation of epithelial characteristics such as the expression of EpCAM, keratins, and E-cadherin, while concurrently upregulating the activity of matrix metalloproteinases (MMPs) [43]. This orchestrated adaptation empowers the cells to navigate through the local extracellular matrix (ECM) and gain access to the microvasculature [44]. The modulation of these molecular processes is integral to CTCs’ ability to overcome the numerous obstacles presented within the bloodstream, contributing to their survival and potential for metastatic dissemination [36].

A pivotal stage involves the endurance of CTCs within the bloodstream [45]. This process encompasses not only withstanding the mechanical stresses within blood vessels but also evading detection by the immune system, which is constantly surveilled by a multitude of blood cells [46]. This is evident in the observation that blood from individuals with cancer contains fragments of cell DNA released from tumor cells [47,48]. In the phase of evasion, CTCs receive crucial support from non-tumor cells. Noteworthy observations of tumor cell morphology arrested in capillaries have illuminated a close association between tumor cells and activated platelets [27].

Platelets exhibit remarkable efficiency in enveloping CTCs, providing a shield against formidable shear forces. This orchestrated interaction is pivotal for tumor cell-induced platelet aggregation, facilitating the processes of extravasation and adhesion [49]. They contribute significantly to immune evasion by releasing transforming growth factor-β (TGF-β), a key factor in deactivating NK cells [44]. TGFβ1 played a key role in switching cells from collective to single-cell motility by orchestrating a transcriptional program involving various proteins [50,51]. Blocking TGFβ signaling resulted in the exclusive observation of collective migration [52]. Additionally, the transfer of the major histocompatibility complex class I (MHC I) complex from granular platelets to CTCs serves as a protective shield, effectively safeguarding CTCs from the cytotoxic attacks launched by NK cells [53]. This mechanism underscores the multifaceted strategies employed by CTCs in collaboration with platelets during the critical escape phase [54].

### 2.3. The Role of CTCs in Metastatic Process

The characteristics of CTCs could undergo changes due to biochemical regulatory adjustments and engagements with blood constituents while navigating through the microcirculation. These interactions induce modifications in cellular deformability and rigidity [55,56]. Moreover, the engagements between CTCs and hematopoietic cells, as well as stromal cells, are vital in coping with physical stresses and maintaining stability during the arrest phase [57]. Our understanding of epithelial cancer metastasis primarily originates from mouse models, delineating a sequence of steps: the EMT of individual cells within the primary tumor, leading to their intravasation into the bloodstream; the survival of these CTCs within the bloodstream; and ultimately, their extravasation at distant sites [58]. At these distant locations, a mesenchymal-to-epithelial transition (MET) occurs, culminating in the proliferation of CTCs as epithelial metastatic deposits [59]. The physical attributes of both individual CTCs and CTC clusters are recognized as significant factors influencing metastatic potential, particularly in their ability to withstand challenges posed by the loss of cell adherence and shear forces within the bloodstream [60]. It is noteworthy that clusters of CTCs or tumor microemboli are suggested to possess a higher metastatic potential than single CTCs [61,62,63]. Compared to single CTCs, aggregations of carcinoma cells offer survival advantages [64]. CTCs within clusters express higher levels of cell adhesion molecules, allowing them to anchor to neighboring cells and escape cell death by anoikis [63,64].

Furthermore, the formation of clusters provides protection against various stresses, including shear stress and immune surveillance, enhancing their survival capabilities [63,65]. Larger cellular clusters are more likely to be entrapped in narrow vasculature, leading to more efficient colonization at new sites and less time spent in circulation [55]. The presence of CTC clusters in circulation correlates with poor prognosis in metastatic BC (MBC) patients [21]. Interestingly, clustered CTCs exhibit a characteristic DNA-hypomethylation pattern associated with increased proliferation and an enhanced stemness phenotype [66]. This methylation signature is indicative of a poorer outcome in BC patients [64]. While the biology of CTCs is not fully elucidated, there is growing evidence for their clinical utility as both diagnostic markers of potential metastasis and prognostic markers of outcome [67]. Consequently, the metastatic capability of isolated CTCs could be constrained, and further investigation concentrating on CTC clusters rather than individual cells may pave the way for innovative treatment approaches [68].

## 3. Diagnosis of CTCs in Breast Cancer: Detection Methods, Limitations, and Precision in Detection

Following enrichment, an identification step is necessary to use immunological, molecular, or functional techniques to detect CTCs encircled by leftover leukocytes at the single cell level [69]. Predominantly, these methods utilize antibodies against various membrane and cytoplasmic antigens, encompassing epithelial, mesenchymal, histospecific, and tumor-related markers for direct immunological detection [44] (Table 1). The variations in target antigens may stem from biological characteristics, such as different protein marker expressions or physical characteristics including size, density, deformability, or electric charges [70]. The combination of these enrichment principles can be fine-tuned to maximize the yield of CTCs (Table 2) [71].

The isolation of CTCs is currently obtained by considering certain parameters as well as exploiting their unique properties. Their larger diameter than that of other blood cells renders to their isolation using size-based isolation techniques. Current methodologies typically involve the following two-step process: initial cell enrichment followed by subsequent detection. Enrichment methods focus on isolating CTCs from the complex blood sample, whereas detection methods aim to identify and characterize these isolated CTCs [24]. Enrichment methods exploit the unique properties of CTCs, such as their larger diameter compared to other blood cells, to facilitate their isolation [44]. Straightforward size-based approaches for CTC capture involve methods, such as isolation by size of epithelial tumor cells (ISET), which combine enrichment and detection steps [72]. Overall, without any prior immune system-based selection, ISET technology enables the extraction of circulating tumor microemboli (CTM) and CTCs from various malignancies as intact cells [73]. Various commercial devices, kits, and reagents are accessible for this purpose, such as the Ficoll-Hypaque and the OncoQuick system from Hexal, utilize centrifugation in tubes equipped with a porous barrier and a medium to establish an appropriate density gradient for CTC isolation [74]. It is crucial to understand that these techniques are mainly employed to separate CTCs from other cells and remove unwanted elements, rather than for identifying them. For identification purposes, alternative instruments such as the DEPArray are necessary [44].

However, it is important to acknowledge that the sensitivity of the ISET technique for detecting smaller CTCs (less than 8 μm) may be limited, potentially leading to false negatives in certain cases [44]. Filters containing captured CTCs from these devices can be positioned on conventional glass microscopy slides for cytological analyses. Alternatively, they can be placed in multi-well tissue culture plates or tubes for the extraction of nucleic acids or proteins [75]. This technology has several benefits, such as the ability to easily study tumor cells recovered onto the filter using methods linked to cellular and molecular biology, as well as techniques connected to the detection and identification of CTCs and their hidden gene abnormalities [76]. Notably, ScreenCell demonstrated a 55% recovery rate and 100% specificity in blood samples that were artificially enriched with the MDA-MB-231 BC cell line [77].

Following enrichment, detection methods employ various techniques to identify and characterize isolated CTCs. Positive selection detection techniques, such as the CellSearch system, Adnatest system, MagSweeperTM, CTC Chips, Herringbone Chips, EPHESIA CTC Chips, IsoFlux, Velcro-like devices, GEDI microdevices, and DEPArray, utilize specific markers to detect CTCs [78]. For instance, the CellSearch platform, the only FDA-approved clinical application for CTC detection, employs fluorescently labeled antibodies to epithelial cytokeratin (CK) as CTC markers, while CD45 staining excludes leukocytes [44,79]. However, limitations exist, such as reliance on the expression of EpCAM and low sensitivity for CTC detection (one cell per 1 mL of blood sample), low purity of isolated CTCs due to potential contamination from leukocytes, and high costs associated with equipment and reagents, limiting widespread adoption [80]. Alternatively, centrifugation using a density gradient isolates CTCs based on the distinct density of leukocytes, red blood cells, and cancer cells [81]. Other drawbacks of the technique include limited specificity in isolating CTCs due to the possibility of co-isolation of other blood components with similar densities, variable recovery rates based on sample characteristics, which can lead to inconsistent results, and the need for specialized equipment and expertise, which can increase the complexity of the procedure [82].

Manual isolation of identified CTCs is laborious, prompting an alternative automated approach using the DEPArray, a device employing dielectrophoresis (DEP) for trapping single CTCs in DEP cages [83]. DEP is a liquid biopsy separation assay based on the differential movement of particles under a non-uniform electric field. DEP represents a relatively recent and continuously advancing technique for isolating CTCs by leveraging their dielectric properties [70]. The dielectric properties of a cell, particularly its polarizability, are contingent on factors such as its diameter, membrane area, density, conductivity, and volume. DEPArray offers the following significant advantage: it allows the recovery of viable CTCs, facilitating RNA sequencing at the single-cell level, which is not feasible with fixed cells [84,85]. Additionally, this platform enables the retrieval of viable cells suitable for culture. Numerous studies have evidenced the successful culturing of CTCs post-recovery through the DEPArray platform [86,87,88]. DEP can be integrated with field-flow fractionation (FFF), a process in which cells are introduced into a chamber and exposed to both an alternating electric field and a meticulously regulated hydrodynamic flow [89]. The combination, known as DEP-FFF, has demonstrated the capability to detect a single tumor cell among 105 peripheral blood mononuclear cells [76]. Notably, this method does not necessitate cell labeling, and it enables the capture of viable cells that can be isolated and cultured. However, challenges include low sample volumes and variable dielectric features due to ion leakage [90].

Flow cytometry (FC) facilitates the quantification of surface and intracellular antigens in individual cells, allowing the detection of specific cell types. This is achieved by utilizing monoclonal antibodies conjugated with fluorescent dyes [91]. In the context of CTC detection, the most targeted antigens are cytoskeletal proteins and cytokeratins, yet it is essential to consider potential limitations, such as limited specificity. A fluorescence-activated cell sorting (FACS) approach was employed for CTC detection and phenotypic analysis, necessitating a pre-enrichment step [92]. FACS involves immunomagnetically enriched blood samples, laser interrogation, and subsequent sorting based on light scattering and fluorescence patterns [93]. While FC offers benefits in terms of surface and intracellular antigen quantification, drawbacks such as high detection costs and the potential for fixed or lysed cells during the assay process should be noted. In addition, reverse transcription polymerase chain reaction (RT-PCR) analysis has been widely used as one of the most frequently utilized methods for identifying CTCs, while combining more than two specific markers (EpCAM, CK19, and hMAM) has proven to be even more effective [94]. Detection at the mRNA or DNA level involves PCR tests with specific primers for tissue-specific transcriptions or tumor-specific mutations, translocations, or methylation patterns. RT-PCR assays are user-friendly, but digital droplet PCR (ddPCR) offers advantages in terms of cost and sample preservation [95,96].

Immunology-based technology frequently utilizes distinct protein biomarkers expressed exclusively by either cancer cells or blood cells, along with their corresponding antibodies. Although this method targets CTCs with specificity, it could be constrained by the availability of certain biomarkers and the possibility of variations in antigen expression among other CTC subpopulations [97]. The enrichment of CTCs through the magnetic-activated cell sorting system (MACS) involves tagging CTCs with superparamagnetic MACS MicroBeads coated with antibodies specific to surface antigens on CTCs [89]. MACS has benefits in terms of efficiency and simplicity, but it may have low purity because of the non-specific bead binding to other cell types and the loss of uncommon CTCs during the enrichment procedure [76]. The separation process involves passing samples through an MACS Column within a MACS Separator containing a powerful permanent magnet [98]. This magnetic field causes labeled cells to be retained, while unlabeled cells pass through unhindered [99]. Additionally, the Epithelial Immunospot (EPISPOT) assay was introduced as a method for detecting viable CTCs in cancer patients. In this technique, proteins that are secreted, shed, or released are immunocaptured on the membrane during short-term cultures. It could be constrained by the requirement for quick cultures, which might have an impact on the assay’s sensitivity and accuracy [90].

Additionally, functional assays, such as the EPISPOT assay, provide quantitative and qualitative information about viable CTCs based on the fluorescence detection of specific epithelial proteins [70]. The EPIDROP, a more rapid and sensitive version, allows for single cell imprinting and discrimination between viable and apoptotic CTCs, offering potential for further molecular characterization. Nonetheless, challenges may arise in standardizing protocols and interpreting results due to the complex nature of single-cell analysis [100]. These tests might not fully reflect the variety of CTC populations, though, since they depend too much on certain protein markers [101]. Moreover, Immunophenotyping with antibodies to specific proteins remains a common approach but is limited to a few proteins of interest. A micro-fluid single-cell western blot (scWB) technology has been developed for proteomic CTC phenotyping but is limited to evaluating only eight proteins. Though it may be used for high-throughput analysis, this approach might not provide the same depth of proteome profiling as other methods, including mass spectrometry [102]. Furthermore, in vitro cultures of CTCs, despite challenges, offer insights into drug testing and the molecular characteristics of metastasizing CTCs. Nevertheless, the transfer of findings to in vivo circumstances may be limited by difficulties including preserving cell viability and reproducing the tumor microenvironment in culture settings [103].

Advancements in the characterization of CTCs involve innovative technologies for personalized cancer treatment. A highly effective microfluidic device, named CTC-chip, was demonstrated to capture EpCAM+ cells using antibody-coated microposts [89,104]. The CTC-chip stands out for its utilization of whole blood without any preprocessing. Due to the minimal shear stress experienced by CTCs during their passage through the chip, an impressive 98% of the captured cells are able to maintain viability [105]. Moreover, cytogenetic analyses, such as fluorescence in situ hybridization (FISH), can identify chromosomal rearrangements. Single-cell analysis, employing array comparative genome hybridization (array CGH) or next-generation sequencing (NGS), allows for genome-wide assessments of duplicate number aberrations and specific mutations, offering valuable insights into cancer heterogeneity [106,107].

Analyzing and identifying CTCs in the peripheral blood could offer crucial prognostic information and might help to monitor the effectiveness of therapy [44]. Challenges such as low cell recovery, poor purity, and diminished viability occur while using enrichment devices that utilize the physical and biological properties of CTCs. Emerging technologies, such as the CTC-chip, demonstrate high viability maintenance, capturing EpCAM+ cells directly from whole blood without preprocessing [23,90]. In conclusion, each technique possesses distinct qualities and continuous investigation endeavors to tackle obstacles to augment their therapeutic effectiveness. Notwithstanding existing constraints, the constant incorporation of these technologies has the potential to enhance our comprehension of cancer heterogeneity and provide guidance for individualized therapy approaches.

**Table 2 ijms-25-04832-t002:** CTC enrichment and detection techniques: This table exhibits the variety of techniques that are used for CTC isolation, enrichment, and detection. The advantages and the limitations of each method are also revealed to offer a better understanding of the importance of ongoing research for more effective techniques.

	Name	Commercially Available Providers	Method	Antibodies	Advantages	Disadvantages	References
Morphology-based enrichment techniques	Isolation by size of epithelial tumor cells (ISET)	Screen Cell (Screen Cell, Paris, France), CTCBIOPSY^®^ (YZYBIO Company, Wuhan, China)	Size-based filtration by using a polycarbonate membrane with 8 μM cylindrical pores	-	Easy, fast affordable, high sensitivity, compatible with many cancer types	Results may be impacted by the morphological variability of CTCs	[73]
	Density gradient	Ficoll-Hypaque (Cytiva, Marlborough, MA, USA), OncoQuick Assay (Hexal Gentech, Holzkirchen, Germany)	Density gradient centrifugation	-	Cell viability, low cost, fast	Low sensitivity	[74,81]
	Dielectrophoretic field- flow fractionation (DEP-FFF)	ApoStream^®^ (Apocell company, Houston, TX, USA)	Separation based on the dielectric characteristics of CTCs combined with field-flow fractionation	-	Potential to acquire viable cells for isolation and cultivation, brief processing time	Low sample volumes and variable dielectric features due to ion leakage	[44,108]
Immunology-based enrichment and detection techniques	CTC-Chip	CTC-Chip, CTC-iChip (Massachusetts General Hospital, Boston, MA, USA)	Microfluidic separation on silicon chip microposts with EpCAM antibodies	Cytokeratin	High sensitivity, fast, cell viability	Does not detect CTC clusters	[105,109]
	Magnetic-activated cell sorting (MACS)	MACS (Miltenyi Biotec, San Jose, CA, USA)	Capture by immuno-labeled magnetic microbeads using superparamagnetic nanoparticles and columns	Cytokeratin, EpCAM, EGFR, and HER2	High sensitivity, automated isolation	Expensive	[99,110]
	CellSearch System	CellSearch (Menarini Silicon Biosystems, Castel Maggiore, Italy)	Immuno-magnetic separation, Flow cytometry and immunofluorescence imaging	EpCAM, CKs 8, 18, 19	High sensitivity, specificity, and reproducibility	Low sensitivity for cells with low EpCAM expression	[109,110,111,112,113]
	AdnaTest	AdnaTest (Qiagen, Hilden, Germany)	Immunomagnetic separation and multiplex RT-PCR	MUC-1, HER-2, EpCAM, CEA, EGFR, PSA, PSMA, Aldehyde dehydrogenase 1 (ALDH1)	Specific enrichment and high sensitivity	Long processing time, expensive	[112]
	EPithelial ImmunoSPOT assay (EPISPOT)	-	Negative selection using anti-CD45 immuno-magnetic beads to capture secreted protein of interest	Cathepsin D, MUC1, CK19, PSA	Detects only viable cells	Protein used must be actively released from cells	[44,70]
	ieSCI-chip multilayer microfluidic system	-	Label-free CTC isolation, CTC enrichment, and single-cell immunoblotting (scWB)	Surface, intracellular, and intranuclear proteins at single-cell resolution	Effective isolation, significant enrichment, and direct molecular functional protein characterization of individual rare CTCs at the single-cell level	Limited to evaluating only eight proteins	[114,115]
Cytometry-based detection techniques	Flow cytometry (FC) and NGS	-	Measurement of surface and intracellular antigens utilizing antibodies linked to fluorescent dyes combined with NGS	Depends on the proteins expressed in primary tumor	Provides information on the genotype and phenotype of single cells	Low sensitivity, time-consuming	[116,117]
	DEPArray (Dielectrophoresis-based isolation)	DEPArray NxT (Menarini Silicon Biosystems, Bologna, Italy)	Dielectrophoresis-based detection and recovery using a microfluidic platform	-	Precise isolation and recovery of individual CTCs based on dielectric properties; enables single-cell analysis and characterization; provides high-resolution imaging for detailed morphological analysis	Not an enrichment technique; CTCs must be enriched before loading onto the DEPArray; limited throughput compared to some other detection methods	[87]

## 4. Immunological Profile of Breast Cancer Patients and CTCs

### 4.1. The Complex Interplay between Immune Dynamics and CTCs in Breast Cancer

Current research has demonstrated that the immune system plays a dual role in BC, as it has the ability to either inhibit tumor growth through immune surveillance or inadvertently promote tumor progression by fostering an immunosuppressive microenvironment [117]. The balance among these opposing forces hinges on the delicate interplay between immune cells and cancer cells, making the immunological profile a critical determinant of disease outcome [118]. The presence of CTCs, particularly those with a positive status (CTCs+), has been linked to metastasis and poor prognosis in BC patients. Recent research has highlighted the active role of the immune system in modulating the fate of CTCs [29,119]. During the initial stages of metastasis, various immune cells interact with CTCs, including neutrophils, NK cells, monocytes, macrophages, and T lymphocytes [120]. While the role of systemic immunity in cancer patients has been extensively explored in terms of clinical significance, the relationship between systemic immunity and CTCs remains unclear [121,122].

Immunological factors, such as tumor-infiltrating lymphocytes (TILs), the expression of immune checkpoint molecules, and the overall immune response, contribute to the classification and staging of cancer [123]. T-cell activation initiates through the recognition of peptide epitopes presented on MHC molecules via the T-cell receptor (TCR) [110]. Recent studies across various cancer types indicate that assessing TCR diversity, clonality, and dynamic changes in circulating T-cells during therapy serves as a valuable tool to estimate anti-tumor activity, define the host-tumor interaction, and predict therapy response [124]. Tumors with high TIL levels often indicate a more robust immune response against cancer cells [125]. In BC patients, the presence of CTCs has been associated with a reduction in CD3+, CD4+, and CD8+ T cells [121]. On the other hand, CTCs express “immune decoy receptors” facilitating immune evasion, including programmed death 1 (PD-1) and its ligand (PD-L1), crucial checkpoint proteins in regulating the anti-tumor immune response [126]. The expression of immune checkpoint molecules is used to predict responses to immunotherapy. High PD-L1 expression may suggest that a patient could benefit from immune checkpoint inhibitors, impacting both diagnosis and treatment decisions [124]. CTCs may also interact with CD4+ Treg cells, contributing to defects in T-cell adaptive immunity and actively suppressing immune function [126] (Figure 1).

Neutrophils, traditionally seen as the first responders to inflammation, unveil a dual role in BC [127]. Their increased circulation is associated with a poorer prognosis, underscoring their potential as both contributors and indicators of cancer progression [128]. Neutrophils, forming clusters with CTCs through VCAM-1-dependent intercellular junctions, express genes that propel cell-cycle progression [34]. Interactions with neutrophils stimulate different gene expression profiles, promoting more intensive metastasis [129]. This interaction occurs within the primary tumor microenvironment before CTC detachment into the bloodstream. Similar cluster formations between CTCs and polymorphonuclear myeloid-derived suppressor cells (PMN-MDSC) have been observed, wherein MDSC infiltrates encourage tumor growth and suppress cytotoxic T-cell activity [120]. The interaction between neutrophils and CTCs involves cell–cell junctions and adhesion proteins, such as cadherin, integrin, and surface glycoprotein. Neutrophils, through the release of interleukin-8 (IL-8), contribute to CTC sequestration and extravasation behaviors, facilitating metastasis [130]. Additionally, the condition known as NETosis, wherein neutrophils release neutrophil extracellular traps (NETs) composed of DNA, histones, and antimicrobial proteins, has emerged as a critical process in cancer progression [112].

Key immune components involved in this phase include tumor-associated macrophages (TAMs), myeloid-derived suppressor cells (MDSCs), neutrophils, and platelets. In many solid malignancies, TAMs often exhibit a tumor-promoting phenotype with impaired phagocytic functions [111]. Targeting the “do not eat me” signal, CD47, through monoclonal antibodies (mAbs), has been shown to reverse CTC immune evasion [131]. Wang et al. demonstrated that the simultaneous blockade of PD-L1 and CD47 in a murine BC model more effectively reduces metastasis compared to single therapy by inhibiting CTCs [132]. Although the specific mechanism behind the reduction in CTCs was not extensively explored, other studies have indicated that CD47 blockade enhances macrophage-dependent phagocytosis [133,134]. Recent research has proposed EpCAM as a potential target antigen for Chimeric Antigen Receptor T-cell (CAR-T) therapy, offering a selective approach to eliminate CTCs. However, reported efficacy has been accompanied by an unfavorable toxicity profile [135]. In the initial phase of BC, the immune response tends to be more effective in identifying and eliminating CTCs [130].

NK cells, a crucial component of innate immunity, participate in the early detection and destruction of CTCs. NK cells recognize cells lacking MHC-I markers, a feature often displayed by CTCs, marking them for apoptosis [129]. NK cells induce tumor cell lysis by secreting tumor necrosis factor-related apoptosis-inducing ligand (TRAIL), binding to death receptors on the cancer cell surface [126]. Reduced NK cell numbers and activity correlate with the presence of CTCs, emphasizing the intricate link between immune surveillance and CTC evasion [130]. Additionally, CD8+ T cells, known for their cytotoxic capabilities, contribute to the immune response against CTCs, aiming to thwart their metastatic potential [120].

Moreover, platelets play a crucial role in the metastasis and progression of cancer. They form aggregates with CTCs, supporting survival, seeding, and outgrowth at secondary sites [126]. Platelets contribute by releasing growth factors, such as PDGF, EGF, and VEGF, inducing tumor angiogenesis and enhancing blood vessel permeability through the release of MMPs, 5-hydroxytryptamine, and histamine [136]. MDSCs, TAMs, and neutrophils produce various proteases, including matrix metalloproteinase 9 (MMP-9), fostering matrix digestion and remodeling to facilitate tumor cell migration and extravasation into blood vessels [137]. Furthermore, platelets and neutrophils actively promote CTC adhesion to endothelial cells [138].

Myeloid-derived suppressor cells (MDSCs) are implicated in immunosuppression and metastatic dissemination [139]. CTC-MDSC clusters may evade T cell responses, contributing to immune evasion. MDSCs secrete proinflammatory factors and endothelial growth factors to instigate tumor angiogenesis [140]. Additionally, MDSCs release IL-6, TGF-β, EGF, and HFG, promoting EMT in tumor cells [137]. Cancer-associated fibroblasts (CAFs), abundant in the tumor microenvironment, play a significant role in tumor initiation, angiogenesis, and metastasis. CAFs can be transferred by CTCs from original tumors to metastatic locations, enhancing tumor cell survival and promoting metastasis [141].

In MBC, where CTCs are more abundant and diverse, the immune landscape becomes more intricate. CTCs form clusters with immune cells, such as neutrophils, potentially influencing their pro-tumor or anti-tumor functions [142]. Neutrophils, initially considered passive players, can exhibit both pro-tumoral and anti-tumoral activities [128]. CTCs undergo phenotypic changes, such as EMT, suggesting the potential development of varied immune escape mechanisms in CTCs compared to primary tumor cells [34]. In MBC, CD47 is expressed in most CTCs and is considered one of the markers identifying CTCs with tumor-initiating capacity. A recent report identified PD-L1-positive CTCs in patients with HR+/HER2−MBC [143].

Immunotherapy seeks to harness the body’s immune system to combat cancer, with CTCs serving as potential indicators of metastasis [117]. Immunological factors influencing CTC dynamics include immune surveillance, evasion mechanisms employed by CTCs, and the overall immune competency of the patient [144]. Therapeutic approaches, such as immune checkpoint inhibitors (ICIs), adoptive cell therapies, and therapeutic vaccines, aim to bolster the immune response against cancer cells, including CTCs [145]. The success of immunotherapy against CTCs hinges on factors, such as CTC immunogenicity, antigen expression, and the immune system’s capacity to recognize and eliminate these cells [146]. The TME, replete with immune-suppressive elements, further modulates the efficacy of immunotherapies against CTCs [147]. In the oncology landscape, immunotherapy’s transformative impact is evident through ICIs, such as ipilimumab, nivolumab, and pembrolizumab [145]. These drugs target proteins such as CTLA-4 and PD-1, crucial for downregulating immune cell activation [148]. Recent studies revealing PD-L1 expression on CTCs offer a non-invasive means to assess PD-L1 status in real-time, presenting CTCs as potential biomarkers in the context of immunotherapy [149]. A compelling case in MBC showcased dynamic changes in PD-L1 positive CTC proportions during combination immunotherapy (nivolumab/ipilimumab), highlighting the potential of integrating CTC analysis to advance personalized cancer treatment strategies [150].

### 4.2. Immunological Dynamics across Breast Cancer Subtypes: A Comprehensive Exploration of CTCs and Immune Responses

The relationship between the immunological profile and CTCs in BC is highly subtype specific. Each subtype presents a distinct interplay with the immune system, influencing the abundance and behavior of CTCs [151,152]. Tailoring therapeutic approaches based on the intricate dynamics between the immune response and CTCs in Luminal A, Luminal B, TNBC, and HER2+BC is essential for optimizing patient outcomes [153]. Firstly, in the Luminal A BC subtype, characterized by the expression of ER and/or PR and low levels of the proliferation marker Ki-67, the immunological profile is often associated with a more favorable response [154]. Luminal A tumors tend to have lower immune infiltration compared to other subtypes, and the presence of CTCs in the bloodstream may be relatively lower [155]. The immunological response against Luminal A CTCs primarily involves ER-related mechanisms, with hormonal therapies targeting these receptors playing a crucial role in managing the disease [156]. On the other hand, Luminal B BC, a subtype with higher proliferative activity and increased expression of Ki-67, demonstrates a more complex relationship with the immune system [157]. These tumors may exhibit variable levels of immune cell infiltration, influencing the interaction with CTCs. The immunological profile of Luminal B BC is often nuanced, incorporating both hormonal and immune-related factors in the management of CTCs (Figure 2) [152].

Triple negative breast cancer (TNBC) poses a formidable challenge in the realm of oncology due to its aggressive nature and the absence of estrogen, progesterone, and HER2 receptors. In recent research, the relationship between TNBC’s immunological profile and CTCs has come into focus, revealing distinctive features that contribute to the tumor’s aggressiveness [158,159]. TNBC is characterized by a higher likelihood of immune cell infiltration, particularly involving cytotoxic T cells and NK cells [6]. The presence of CTCs in TNBC is associated with a robust immunological response. Notably, targeting immune checkpoints, such as PD-L1, has emerged as a potential therapeutic avenue to enhance the immune response against CTCs in TNBC [160]. A recent systematic review has consolidated current insights on the correlation between CTC numbers and prognosis in metastatic TNBC (mTNBC). The outcomes underscore that a positive CTC status aligns with reduced OS and PFS in individuals diagnosed with mTNBC. Importantly, the decline in CTC numbers during therapy serves as a positive prognostic indicator, emphasizing the significance of CTC detection as a prognostic tool before and during treatment for mTNBC patients [161]. PD-L1 has emerged as a predictive biomarker for the efficacy of ICIs in mTNBC. The landscape of ICIs, including PD-L1, was explored in a cohort of BC patients, revealing a heightened prevalence of the PD-L1+CD45−CK+ phenotype in TNBC compared to Luminal subtypes [162]. Within the TNBC subgroup, this specific CTC phenotype was correlated with significantly reduced OS, highlighting its potential prognostic significance [150].

HER2+ stands out with its distinct immunological landscape, marked by the overexpression of the HER2 receptor [154]. Targeted therapies, such as trastuzumab and pertuzumab, play a pivotal role in shaping the interaction between CTCs and the immune system in HER2+ BC [152]. These therapies enhance the immune response against HER2+ cells, offering a unique approach to treatment. In a study directed by Lim et al., the effectiveness of HER2-targeted therapy combined with chemotherapy was demonstrated in patients with HER2+ primary tumors, irrespective of their baseline CTC count [163]. Particularly noteworthy was the significant reduction in CTC numbers observed when chemotherapy was complemented with HER2-targeting drugs, especially in cases with initially high baseline counts [164]. This finding supports the notion that diverse pathways of tumor progression contribute to clinical heterogeneity and survival outcomes in MBC subtypes [152]. The study proposed the inclusion of CTC count in future therapeutic trials for MBC to refine patient stratification based on prognostic groups [21].

Moreover, in HR+, HER2− BC, investigations into PD-L1 expression on CTCs have revealed a broad range, from 0.2% to 100%, across patients [165]. The presence of PD-L1-positive CTCs has been linked to shorter PFS. Prospective studies in MBC have highlighted the feasibility of detecting PD-L1+ CTCs and their potential as prognostic indicators [150]. Additionally, a study expanded this analysis to include CD47, another key immune checkpoint, revealing its co-expression with PD-L1 on CTCs in MBC patients [149]. The combined high expression of CD47 and PD-L1 on CTCs correlated with disease progression and reduced PFS [161]. Furthermore, Grigoryeva et al. conducted pioneering research uncovering PD-L1 expression on CTCs in individuals diagnosed with HR+/HER2− BC. This discovery suggests a mechanism by which CTCs in this subtype may resist immune attacks [143].

## 5. Clinical Implications and Future Advances

CTCs have emerged as a transformative tool in BC research, offering a non-invasive avenue for comprehensive insights into disease progression, treatment response, and personalized therapeutic interventions [166]. Detecting and diagnosing cancer, especially in its early stages, is vital for clinicians to effectively identify and treat BC patients. This has led to a growing focus on the clinical utility of CTC detection in BC, highlighting its potential significance in the field [5]. Technological developments have also added to the clinical importance of CTC detection. The accuracy and dependability of CTC enumeration have increased due to the development of more precise and sensitive procedures, such as the use of microfluidic devices and sophisticated imaging techniques [167,168]. In addition to increasing detection rates, this technical advancement has made it possible to characterize CTCs at the molecular level, providing important information on the heterogeneity of tumor cells and possible treatment targets [169].

Considering CTCs as prognostic and predictive biomarkers, their clinical importance in BC is highlighted. Investigations, such as the seminal work by Banys-Paluchowski et al. and Dijkstra et al., have established a robust link between the detection of CTCs and an elevated association between disease recurrence and metastasis [144,170]. This demonstrates how CTCs might assist physicians in figuring out the level of disease aggression and adjusting therapies accordingly. The accuracy of CTC enumeration has increased due to ongoing advancements in detecting technologies, such as microfluidic devices, giving physicians an invaluable tool for real-time monitoring [76]. In BC, targeting ER, PR, and HER2 has been a cornerstone of therapeutic strategies. However, the discordance in HER2 status between primary and metastatic tumors presents a challenge [171]. By enabling the identification of high-risk patients and starting anti-cluster treatment early in the course of the disease, CTCs provide a unique perspective that may help to lower the chance of metastasis. Targeted HER2 therapy has also shown promise in lowering the overall count of CTCs, illustrating the interaction between targeted treatments and CTC dynamics [18,150].

While CTCs are primarily found as single cells, a subset exists in the form of clusters, which can enhance their survival in the bloodstream and are linked to increased metastatic potential [37,61]. Although the clinical implications of CTC clustering in early BC remain unknown, considering the recognized metastatic potential of CTC clusters, it is hypothesized that they may contribute to the risk of disease recurrence. In the context of MBC, the presence of CTC clusters has been associated with a poorer prognosis and reduced survival [172]. Therapeutic strategies are being developed to inhibit CTC cluster formation and disrupt existing clusters into single cells, aiming to mitigate their metastatic capacity. Na+/K+-ATPase inhibitors, for instance, have demonstrated the ability to dissociate CTC clusters, and an ongoing phase I clinical trial is investigating the use of digoxin in advanced BC [14,138]. One of the most exciting aspects of CTC research lies in its potential contribution to personalized medicine [144]. Ongoing clinical trials are actively exploring the prognostic value of specific molecular alterations in CTCs, as detailed in the Table 2.

The investigation of novel strategies to target CTCs is indicative of how active BC research is. Precision medicine’s future is shown by techniques like the usage of selectin-based implanted shunt devices adorned with E-selectin molecules and tumor necrosis factor-related apoptosis-inducing ligand (TRAIL) [47,173]. Even though TRAIL treatment is limited in solid tumors, a promising approach is the possibility of increased sensitivity following the separation of TRAIL-resistant cancer cells from the extracellular matrix [174]. Furthermore, a major advancement has been made with the creation of pre-clinical models obtained from CTCs, such as three-dimensional organoids. These models closely resemble the behavior and heterogeneity of cancer cells within the tumor mass, in addition to providing stable shape, gene expression, and cell signaling [175]. This breakthrough has enormous promise for personalized therapies and treatment development. Additionally, employing organoids to study human cancer, heterogeneity, and metastasis has made genome editing techniques—more especially, CRISPR/Cas9—essential. Since 2012, the fusion of CRISPR with organoids has yielded significant discoveries in the field of molecular biology [176,177]. Patient-derived xenograft models (PDX) also contribute to understanding the properties of CTCs required for metastasis and testing potential anticancer drugs. However, the development of PDX models is time-consuming, limiting their immediate applicability in clinical decision-making [178].

As detailed in the previous section, cancer immunotherapies, including CTLA-4 and PD-1/PD-L1 inhibitors, have shown substantial clinical benefits [150,179]. An interesting study established a platform for culturing autologous tumor organoids with peripheral blood lymphocytes to evaluate and stimulate tumor-specific T cell responses to epithelial cancers. This approach presents a viable option for patients with advanced cancer by isolating tumor-reactive T cells and evaluating the therapeutic impact of T-cell-mediated assaults [180]. Thus, several enrichment strategies have been used to address the problem of CTC rarity in the circulation. One noteworthy technique for identifying CTCs from blood samples is the FDA-approved CellSearch system, which uses immunomagnetic beads coated with antibodies targeting EpCAM [23]. The use of microfluidic technologies has grown in popularity because they offer effective platforms for analysis and separation. Research utilizing these tools has demonstrated potential in forecasting metastasis and directing medical decision-making, particularly for patients with HR+ BC [181].

The development of imaging technology has been essential to improving the accuracy of CTC detection and characterization. CTCs, cell-free DNA, RNA, proteins, and exosomes are all examined in liquid biopsy samples [182]. Clinical research has shed light on CTC heterogeneity and molecular changes, connecting them to therapy response and disease progression [21]. More specifically, research conducted by Banys-Paluchowski et al. has linked the existence of certain molecular changes in CTCs to treatment resistance (Table 3) [144]. The molecular profiling of CTCs has been transformed by NGS, which enables a thorough examination of transcriptomic and genomic changes. Studies by Twomey et al. and Albrecht et al. shed light on the genomic landscape of CTCs, uncovering potential therapeutic targets and mechanisms of resistance [57,183]. The Hydro-Seq technique in BC revealed diverse CTC populations, including those undergoing EMT and MET. The relationship between epithelial/mesenchymal status and HER2 expression highlights the complexity of CTC biology. More specifically, single-cell RNA sequencing (scRNA-seq) has proven pivotal in identifying genes associated with CTC aggressiveness [184]. Studies in BC demonstrated the utility of CTC-derived ER signaling assessed through scRNA-seq for early monitoring of treatment response [57].

Findings from various studies indicate a strong prognostic value of detecting CTCs in BC, spanning different stages of the disease [21]. In mTNBC, a CTC+ status is consistently associated with reduced OS and PFS. These observations lay the groundwork for incorporating CTC counts as crucial clinical markers, guiding treatment decisions and offering a quantitative prognostic indicator for chemotherapy response [185]. The diagnosis of cancer, particularly at an early stage, faces challenges due to the limited presence of CTCs in circulation. To overcome this, microRNA (miRNA) expressions, frequently dysregulated in cancers, were explored as potential tools for early cancer detection. Unique expression patterns or signatures of serum miRNAs in BC have been identified [186]. Interestingly, miRNA panels derived from cancer tissues have shown promise as circulating miRNA signatures, exhibiting consistent expression patterns between tumor tissues and serum samples [187]. Liquid biopsy, comprising CTCs and ctDNA, is becoming increasingly popular as a complete molecular profiling procedure. The incorporation of CTC-based methodologies into standard clinical practice has the potential to improve patient outcomes and provide more efficient, customized treatment of brain cancer. Whether ctDNA levels in MBC patients have a greater predictive influence than CTCs is yet unknown [188]. Despite baseline CTC identification having demonstrated predictive importance, studies on cell-free TP53 mutations in mTNBC did not reveal a significant influence on prognosis. This indicates that the prognostic information offered by ctDNA may be equivalent to or less than that of CTC [144].

Prominent clinical trials, such as TREAT-CTC, SWOG S0500, CirCe01, STIC CTC Trial, CirCe T-DM1, DETECT III, DETECT IV, DETECT V/CHEVENDO, IMENEO Study, SUCCESS-A, STI-CTC III, ECOG-ACRIN E5103, BEVERLY-2, and others presented in Table 3 cover a spectrum of BC stages, including neoadjuvant and metastatic phases [161,165,189,190,191,192]. These investigations scrutinize the value of detecting and quantifying CTCs as prognostic and predictive indicators. Their findings shed light on the role of CTCs in forecasting outcomes such as DFS, OS, and response to treatment. Moreover, these trials explore the feasibility of tailoring therapeutic approaches based on CTC detection, aiming to enhance patient care and therapeutic outcomes [161,193,194,195,196].
ijms-25-04832-t003_Table 3Table 3“A Comprehensive Overview of studies Investigating Circulating Tumor Cells (CTCs) in Diagnosis and Prognosis”: This table presents a comprehensive overview of diverse studies exploring the role of CTCs in breast cancer (BC), encompassing various study types. More specifically it includes clinical trials, meta-analyses, prospective trials, neoadjuvant studies, retrospective analyses, ongoing trials, studies on drug resistance, investigations into CTC clusters, correlations between CTCs and therapy response, and assessments of long-term impact. The studies contribute valuable insights into CTCs’ prognostic significance, therapeutic implications, and their potential role in shaping personalized treatment approaches for BC patients across different stages and subtypes.Study(Reference)Number of PatientsStageSubtype of BCKey Findings/ResultsTREAT-CTC (NCT01548677)[180]1317Neoadjuvant MBCHER2− EBCHalted for futility; no significant difference in disease-free survival (DFS) after additional trastuzumab or observation; failure attributed to inappropriate treatment intervention.SWOG S0500 (NCT00382018)[181]595MBCNot specifiedEarly treatment change based on elevated baseline CTCs did not show significant survival improvement; reaffirmed CTCs’ prognostic impact.CirCe01 (NCT01349842)[182]265MBCNot specifiedInvestigated CTC decrease after one chemotherapy cycle in MBC patients; compared conventional assessments with CTC-driven intervention.STIC CTC Trial (NCT01710605)[183]755MBCHER2−, HR+ MBCNon-inferior progression-free survival (PFS) for CTC-driven decisions; potential benefits in certain subgroups.CirCe T-DM1 (NCT01975142)[184]105MBCHER2− MBCLimited efficacy of trastuzumab emtansine (T-DM1) in HER2− MBC; early closure of the trial.DETECT III (NCT01619111)[185]105OngoingMBCHER2+ MBCRandomizes patients with HER2+ CTCs to standard endocrine or chemotherapeutic treatment.DETECT IV (NCT02035813)[179]116OngoingMBCHER2− MBCIncludes patients with only HER2− CTCs; treated based on hormonal or chemotherapeutic regimens.DETECT V/CHEVENDO (NCT02344472)[186]270OngoingMBCHER2+, HR− MBCFocuses on HER2+, HR+ MBC; treatment choices not driven by CTCs; aims to develop an “endocrine responsiveness score” based on CTC characteristics.IMENEO Study[189]2156 individual patients from 21 studiesNon-MBCNot specifiedMeta-analysis involving over 2000 non-MBC patients; CTC counts emerged as a strong and independent prognostic indicator for distant-metastasis-free survival, overall survival (OS), and locoregional relapses.SUCCESS-A(NCT 02181101)[178]2000+Adjuvant BCNot specifiedCTC positivity before and after adjuvant chemotherapy independently predicted poor DFS and OS; patients with at least 5 CTCs per 30 mL exhibited the worst prognosis.STI-CTC III(NCT01975142)[183]154MBCER+ HER2− MBCCTC-driven treatment decisions associated with longer progression-free survival (PFS) compared to clinically driven choices. Some limitations due to the lack of standardized clinical criteria for CT in the clinically driven arm.ECOG-ACRIN E5103[186]2859 (386 of African ancestry (AA) and 2473 of European EA)Adjuvant BCNot specifiedCTC detection at two years and five years post-treatment associated with higher risk of death and recurrence; potential benefits of adjuvant radiation therapy for CTC+ patients.SWOG S0500 (Part 2)[181]595OngoingMBCNot specifiedExplores therapy switch options if specific CTC fall thresholds are not met after one treatment cycle; presence of CTCs remains an adverse prognostic factor.DETECT III and IV[188]1933MBCHER2−The existence of one or more CTCs per 7.5 milliliters of peripheral blood, exhibiting robust HER-2 staining, was correlated with a reduced OS.BEVERLY-2[189]135 for CTCsEarly-stage BCHER2+Detection threshold of ≥1 CTCs/7.5 mL before neoadjuvant therapy (NT)predicted shorter DFS.GeparQuattro[176]420(213 and 207 patients before and after NT)Early-stage BCHER2+Detection threshold of ≥2 CTCs/7.5 mL before NT predicted shorter DFS. The identification of CTCs following NT showed no association with DFS or OS.Wei et al.[177]42MBCERα and HER2/neuPresence of multi-drug resistance-related proteins (MRPs) on CTCs predictive of poor response to chemotherapy; correlation with reduced PFS.Schramm et al.[190]105 (14 BC)Not specifiedNot specifiedMRP profiles of CTCs highly predictive of chemotherapy response; correlation irrespective of tumor type and disease stage.Capuozzo et al.[35]119MBCNot specifiedDue to the strong correlation between alterations in CTC counts and pre- and post-therapy imaging outcomes, CTCs can serve as a biomarker that has the potential to predict treatment efficacy at an earlier stage compared to conventional imaging modalities.Gunti et al.[165]16 with CTCsMBCER/PR(+) HER2−Expression of PD-L1 on CTCs in BC patients with ER or PR (+) and HER2−; potential of CTC/PD-L1 assays for liquid biopsy in future clinical experiments focusing on the immune checkpoint in BC patients.Fridrichova et al.[119]16 studies with 2860 BC patients and 1958 controlsNot specifiedNot specifiedMeta-regressions and subgroup analysis examining potential factors contributing to heterogeneity in CTC studies; ongoing work required to enhance the precision of CTC enrichment and detection methods.Magbanua et al.[195]2202MBCNot specifiedCTC enumeration ideal for stratifying stage IV patients; those with indolent disease (<5 CTCs) had longer OS compared to those with aggressive disease (>5 CTCs). Valuable tool for staging advanced disease and patient stratification.


## 6. Discussion

Investigating the molecular pathways through which CTCs facilitate tumor metastasis represents a crucial research avenue. The dynamic nature of CTCs, reflecting the heterogeneity of the primary tumor, poses both opportunities and challenges. Molecular alterations and phenotypic changes in CTCs provide a snapshot of the evolving tumor landscape, offering potential insights into therapeutic targets and mechanisms of resistance. Moreover, the pursuit of CTC cluster targeting presents a promising approach to mitigate the risk of BC metastasis, either by disrupting their collective release or dissociating them within the circulation [37]. This increased potential arises from variations in intercellular adhesion protein expression, DNA methylation levels, anti-apoptotic properties, and immune evasion mechanisms. Consequently, targeting CTC clusters has emerged as an innovative strategy for intervening in BC metastasis [61].

The continuous evolution of diagnostic methods for CTC detection, from immunomagnetic enrichment to cutting-edge technologies, such as microfluidic devices and NGS, showcases remarkable progress. These advancements have significantly enhanced our ability to detect and characterize CTCs; yet, challenges persist. The potential benefits of CTC detection encompass various aspects, including providing independent prognostic information in early BC, monitoring CTCs in advanced BC for predicting therapeutic efficacy, improving patient stratification for (neo)adjuvant therapies, offering molecular insights into specific therapeutic receptors, and exploring stemness and EMT markers in CTCs, a current focal point in cancer research. Exploring immunotherapeutic approaches to diminish or eradicate CTCs represents a novel and viable strategy to impede tumor metastasis or recurrence. However, owing to insufficiently elucidated molecular mechanisms, the development of fully-fledged immunotherapeutic strategies directed at CTCs is currently in its nascent stages [137]. Interestingly, the presence of CTCs and the clinical significance of detecting CTCs differ across subtypes of BC. Interestingly, individuals with metastatic tumors characterized by HR+ and HER2− are more prone to exhibit CTCs in their peripheral blood. However, in cases of HER2+ and TNBC, the available data regarding the prognostic significance of CTCs are inconclusive [152,153].

The aforementioned results indicate that CTCs are subject to both innate and adaptive immune evasion mechanisms, potentially contributing to their metastatic capabilities [31,119]. Currently, the clinical utility of CTCs primarily revolves around predicting treatment response and prognosis based on their enumeration and characterization [143]. The amalgamation of CTC assessment with other biomarkers allows for effective monitoring of disease progression and prognostic predictions [119]. However, the widespread clinical implementation of CTCs is hindered by the associated high costs of isolation and monitoring. Future advancements in CTC detection technology should prioritize achieving enhanced enrichment efficiency and standardization to meet clinical demands. Clinical trials evaluating the impact of CTC detection on the management of cancer patients are ongoing, aiming to determine whether CTCs can serve as predictive biomarkers for therapy response or failure [137,152,187].

## 7. Conclusions

Looking ahead, the integration of CTC assessment into clinical practice holds great promise, particularly in monitoring non-metastatic patients’ post-intervention. Ongoing trials and technological innovations offer glimpses of a future where CTCs contribute to personalized therapeutic strategies. The implications of CTC research extend beyond current knowledge, promising transformative impacts on BC management. The molecular characterization of CTCs is a key focus, aiming to identify novel therapeutics targeting micro metastatic spread and elucidating their connections to cancer stem cells [27,195]. Integrating advanced imaging systems with molecular characterization holds potential to refine prognosis, define treatment strategies, and mitigate the risk of metastasis. Leveraging modern technologies is envisioned to unveil molecular pathways in CTCs and pave the way for designing novel molecular therapies specifically targeting CTCs. Overall, CTCs are emerging as promising tumor biomarkers in BC, offering the prospect of a “liquid biopsy: for patient stratification and real-time therapy monitoring. Researchers anticipate further research to deepen their understanding of the intricate relationships between CTC formation and immune escape, paving the way for more comprehensive and effective immunotherapeutic interventions. The ongoing evolution of CTC research is poised to make a significant impact on personalized medicine, potentially transforming the landscape of BC management.

## Figures and Tables

**Figure 1 ijms-25-04832-f001:**
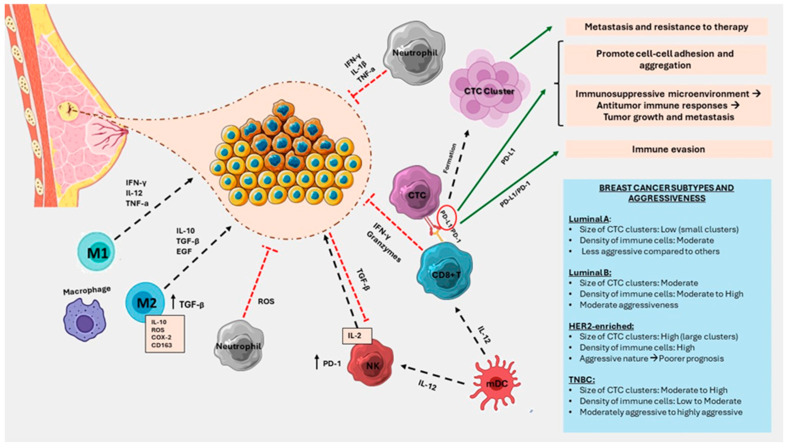
This figure delves into the intricate landscape of the Breast cancer (BC) microenvironment, elucidating the complex interactions between immune components and cancer cells. With a keen focus on subtype aggressiveness, the illustration vividly portrays the dynamic network of interactions driving disease progression. Each subtype’s distinct characteristics are outlined, showcasing their specific patterns of immune cell infiltration and evasion by tumor cells. Furthermore, the figure provides clarity on the cryptic process of circulating tumor cell (CTC) cluster formation, shedding light on their role in metastatic spread.

**Figure 2 ijms-25-04832-f002:**
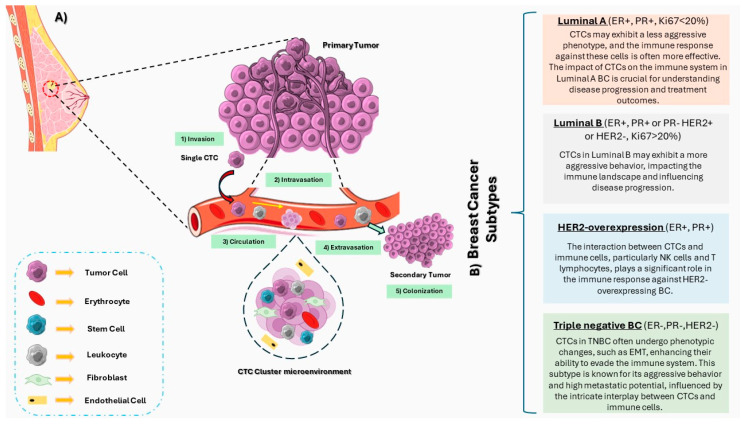
(**A**) This illustrative figure serves as a comprehensive overview of the interactions involving circulating tumor cells (CTCs) within the bloodstream. At its focal point, the graphic vividly depicts a solitary CTC, illustrating its circulation alongside other immune cells. The intricate dance of interactions between CTCs and the immune system is further highlighted, showcasing the dynamic engagement of these cells in the complex environment of the circulatory system. Additionally, the figure delves into the formation and structure of CTC clusters, providing a visual representation of the aggregations that contribute to the overall understanding of CTC dynamics. (**B**) Beyond the central depiction of CTCs, the segmented sections of the figure meticulously unravel the distinct immune dynamics within the context of four prominent breast cancer (BC) subtypes. For triple negative BC (TNBC), Luminal A, Luminal B, and HER2-overexpression, the figure succinctly elucidates the nuanced relationships that shape the impact of CTCs on disease progression. The metastatic process stages are depicted with green boxes, illustrating the key steps involved in metastasis facilitated by CTCs. These stages include intravasation, circulation, extravasation, and colonization, highlighting the role of CTCs in each phase of metastatic progression. Through visually compelling depictions and concise details, the graphic sheds light on how CTCs uniquely influence each BC subtype, offering valuable insights into the intricate landscape of breast cancer and its diverse manifestations.

**Table 1 ijms-25-04832-t001:** This table provides a summary of immunological biomarkers commonly utilized for the detection of circulating tumor cells (CTCs). Biomarkers are categorized by their respective types, including epithelial, mesenchymal, histospecific, and tumor-related markers, along with their corresponding descriptions.

Immunological Biomarker Type	Marker	Description
Epithelial	EpCAM	Epithelial cell adhesion molecule
	Cytokeratins (CK)	Intermediate filament proteins found in epithelial cells
Mesenchymal	Vimentin	Intermediate filament protein found in mesenchymal cells
Histospecific	HER2	Human epidermal growth factor receptor 2
	EGFR	Epidermal growth factor receptor
Tumor-related	CEA	Carcinoembryonic antigen
	CA19-9	Carbohydrate antigen 19-9
	MUC1	Mucin 1
	PSA	Prostate-specific antigen

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
