# Peer review of "Exploring the Immunological Profile in Breast Cancer: Recent Advances in Diagnosis and Prognosis through Circulating Tumor Cells"

_ijms, 2024, doi:10.3390/ijms25094832_

Round 1

Reviewer 1 Report

Comments and Suggestions for Authors

Review of

“Exploring the Immunological Profile in Breast Cancer: Recent Advances in Diagnosis and Prognosis through Circulating Tumor Cells”

Overall comments:

The authors provide a comprehensive review about the immune interaction with CTSs

Specific comments:

Line 37-42 add the incidence percentage for each of the 4 BC subtypes.

What is the difference between DTCs and CTCs? Are CTCs in metastatic target organs named DTCs??

Line 152 what are other genes upregulated in EMT?

Line 172 which “various” proteins are mentioned in refs. 64, 117? Please name them.

Line 174 Please add a reference for the transfer of MHC1 from platelets to CTCs.

Figure 1. What are those stem cells? Please define.

Line 546 replace doctors with physicians

Table 2. The trials should be named and discussed in the text so the reader can directly refer to the original publication.

Comments on the Quality of English Language

Minor editing of English language required

Author Response

Please see the attachment. Thank you, Reviewer 1, for your thoughtful and detailed comments. Your insights have been instrumental in refining our work and ensuring its accuracy and clarity. We appreciate your valuable contributions to this review process.

Reviewer 2 Report

Comments and Suggestions for Authors

The authors provide a very nice summary on the CTC aspect of breast cancer. However, the immunological connection between CTCs and the cancer type is not very clearly explained.  They could explain the connections with a figure if possible or include/modify/extend the current figure. The figure provided should have references for each subtype and the authors should explain terms such as aggressiveness in the figure. Some of the references go to other reviews which may not be relevant such as 170, 173 in line 471 and similar references on line 475. the authors can cite the original publications. 

Author Response

"Please see the attachment." Reviewer 2, your meticulous attention to detail and insightful feedback have significantly contributed to the improvement of our work. We deeply appreciate the time and effort you invested in providing us with valuable suggestions and clarifications. Thank you for your invaluable contribution to this review process.

Reviewer 3 Report

Comments and Suggestions for Authors

Dear Authors,

Thank you for submitting your review in IJMS. Since blood levels of CTCs reflect tumor size and prognosis, they are expected to contribute to personalized cancer medicine, such as monitoring treatment progress. Although CTCs have been shown to have the potential to bring breakthroughs in cancer treatment and diagnosis, they remain largely underutilized in clinical practice. The extremely low blood levels of CTCs and the lack of an established method for their isolation are considered a major problem. This review summarizes the current status of the application of CTCs in the diagnosis and treatment of breast cancer. It is also very interesting to see how the dynamic interaction between CTCs and immune cells affects the progression and metastasis of breast cancer.

The following points should be considered and made more understandable to the readers.

Major points:

1.      References must be numbered in order of appearance in the text (including table captions and figure legends). Please put the numbers of the cited references in order.

2.      There are many verbs with similar meanings in the Abstract, such as emphasize, highlight, underscore, etc., which are difficult to read (they appear 7 times). The authors should work on the text of Abstract more. I think it may be necessary to have English-speaking people look at the paper.

Minor points:

1.      Lines 77-78: “The presence of CTCs in the bloodstream is exceedingly scarce, typically ranging from 1 to 10 CTCs per milliliter of blood”

Lines 128-129: “The quantity of CTCs in the blood is exceedingly limited, ranging from 1 to 10 cells per 10 ml of blood”

Lines 247-248: “one cell per 1 mL of blood sample”

The above statements are inconsistent and confusing to the reader. Please be consistent with the exact numbers.

2.      Line 54: What does TME stand for?

3.      Line 293: EPISPOT instead of ELISPOT

4.      Line 396: MHC-I markers instead of MHC-Imarkers

5.      Line 600: Where do I look in Table 2?

6.      Line 633: What does the (BC) in “the role of CTCs in BC (BC)” mean?

7.   Table 2: What is the HER2-EBC in the "Subtype of BC" column?

8.      Table 2: Reference numbers don't seem to match the references behind the paper.

9.      References: Several identical references appear. For example, references 2 and 3, 6 and 7, 12 and 13, 18 and 19, etc. Please check the references carefully.

Author Response

Reviewer 3 thank you very much for your valuable feedback. "Please see the attachment."

Reviewer 4 Report

Comments and Suggestions for Authors

In this review, Kotsifaki and colleagues attempt to describe the intricate and complex interplay of beast cancer cells and the immunological landscape of this tumor type focusing on CTCs. The authors include a review of the current methods used for enrichment and detection of CTCs, together with some examples of their role in diagnostics and future perspectives. It is a very interesting field of research. However I have some concerns, and some sections must be heavily reviewed to make the manuscript readable. In particular, the part concerning detection and enrichment methods is seriously confusing.

Major comments:

-lines 37-43: please check and rewrite, as it is confused

-line 128: “the quantity of CTCs in the blood is exceedingly limited, ranging from 1 to 10 cells per 10 ml of blood”. Is this sentence valid for all cancer types o some tumor are more prone to spread CTCs (i.e., small cell lunc cancer)?

- paragraph 3 is not clear: while the content is correct, the paragraph must be reorganized entirely since the enrichment and detection methods are confusedly described without a clear distinction. Enrichment methods are different from detection methods. For example: in line 213, the paragraph start talking about the detection steps following enrichment, while in line 239 the authors states that current methodologies involve two steps (enrichment + detection). The latter sentence should be used to introduce the paragraph. Moreover, it is correct to cite the ISET techniques but is should be said that the procedure combine the enrichment+detection steps.

-When the benefits of each technology are reported in the main text, the disadvantages should be described as well. For ISET there are no flows described (line 222)

-line 248: centrifugation using a density gradient such as OncoQuick and Ficoll-like methods are not detection methods but enrichment approaches, as one can exploit them to isolate CTCs (and eliminate undesired cells) and not for their identification, for which other instruments are required (i.e. DEPArray).

-line 267: one advantage of DEPArray is that viable CTC can be recovered allowing for RNA sequencing at single cells level, which would not be possible with fixed cells. Moreover, the authors write that the DEPArray platform allows for the recovery of viable cells that can be cultures. Please cite some articles by literature that describe CTC culturing after DEPArray-based CTC recovery.

-Table 1: the DEPArray NxT is not an enrichment technique but rather a detection method useful also for recovery. CTCs must be enriched before loading on the DEPArray.

-Table 1: the table describes CTC detection and enrichment techniques, so I do not understand why PDX are reported since they are useful for experimental and functional studies and not for enrichment/detection. It is advisable to talk about PDX in the section related to future advances and not here.

-line 376: neutrophils are gaining even more attention in this context, as reviewed here 10.3390/cells13040337. I suggest to cite the condition known as NETosis.  

Minor comments:

-Please check the references: they are not listed in correct numerical order.

-please check the abbreviation as they are not explained at the first use (i.e., ER, PR,ecc).

-in the bibliography, some references are reported twice: i.e. 6=7, 12=13; 18=19, ecc..

Comments on the Quality of English Language

English requires minor editing.

Author Response

Thank you for your diligence in reviewing the text. "Please see the attachment."

Reviewer 5 Report

Comments and Suggestions for Authors

Remarks to the authors:

In this review, the authors have summarized the function and the recent advance of circulating tumor cells. This manuscript is interesting and important to understand the CTC. However, there are some suggestions for authors to improve this manuscript.

1.    In the section of “the role of CTCs in in metastatic process”, a figure that describe this process is important, so authors could insert an illustrative figure in this section.

2.    In the line of 215-217, the authors have mentioned that CTCs enrichment methods mainly rely on the immunological detection, the authors could list a table to summary all the immunological biomarkers.

Comments on the Quality of English Language

Minor English editing is needed.

Author Response

Many thanks for your kind comments and valuable feedback. Your insights are greatly appreciated, and we're grateful for the time and consideration you've dedicated to reviewing our work. Please see the attachment

Round 2

Reviewer 4 Report

Comments and Suggestions for Authors

I would like to thank the authors for the accurate revision and for their correct replies. I suggest the authors to cite these references when they talk about RNA analysis on DEPArray- isolated CTCs, that would further enrich their manuscript:

- 10.1038/s41416-021-01481-z

- 10.3389/fgene.2022.1012191

Author Response

Thank you for your constructive feedback and valuable suggestions. We appreciate your acknowledgment of our revisions and responses.

We have now included the references you suggested (10.1038/s41416-021-01481-z and 10.3389/fgene.2022.1012191) in the relevant sections discussing RNA analysis on DEPArray-isolated CTCs. We also believe this addition significantly enriches the content of our manuscript.

Once again, we sincerely thank you for your time and thorough review.